# In vivo diversification of target genomic sites using processive base deaminase fusions blocked by dCas9

Beatriz Álvarez[1], Mario Mencía[2], Víctor de Lorenzo [3] & Luis Ángel Fernández [1✉]

In vivo mutagenesis systems accelerate directed protein evolution but often show restricted capabilities and deleterious off-site mutations on cells. To overcome these limitations, here we report an in vivo platform to diversify specific DNA segments based on protein fusions between various base deaminases (BD) and the T7 RNA polymerase (T7RNAP) that recognizes a cognate promoter oriented towards the target sequence. Transcriptional elongation of these fusions generates transitions C to T or A to G on both DNA strands and in long DNA segments. To delimit the boundaries of the diversified DNA, the catalytically dead Cas9 (dCas9) is tethered with custom-designed crRNAs as a "roadblock" for BD-T7RNAP elongation. Using this T7-targeted dCas9-limited in vivo mutagenesis (T7-DIVA) system, rapid molecular evolution of the antibiotic resistance gene *TEM-1* is achieved. While the efficiency is demonstrated in *E. coli*, the system can be adapted to a variety of bacterial and eukaryotic hosts.

[1] Department of Microbial Biotechnology, Centro Nacional de Biotecnología, Consejo Superior de Investigaciones Científicas (CNB-CSIC), Darwin 3, Campus UAM Cantoblanco, 28049 Madrid, Spain. [2] Centro de Biología Molecular "Severo Ochoa" (Consejo Superior de Investigaciones Científicas – Universidad Autónoma de Madrid), Nicolas Cabrera 1, Campus UAM Cantoblanco, 28049 Madrid, Spain. [3] Systems Biology Program, Centro Nacional de Biotecnología, Consejo Superior de Investigaciones Científicas (CNB-CSIC), Darwin 3, Campus UAM Cantoblanco, 28049 Madrid, Spain. ✉email: lafdez@cnb.csic.es

D irected evolution enables the selection of protein variants with improved properties[1,2]. The generation of genetic variability followed by a screening process are the essential steps of directed evolution[3,4]. In vitro mutagenesis techniques quickly produce large number of variants of the target gene, but their selection requires, in most cases, cloning and transformation into a host cell for expression. These steps are time-consuming and labor-intensive, especially when iterative cycles of mutagenesis and selection are needed. Cell-free selection methods are also feasible[5], but they are technically demanding and functional expression of complex membrane and oligomeric proteins is difficult to achieve. Hence, in vivo mutagenesis systems are preferred because the generation of variants, their expression, and selection can be done in a continuous process, which accellerates directed evolution[4].

Long-established in vivo mutagenesis methods (e.g., chemical mutagens, radiations, and mutator strains) do not target specific genes and are deleterious for the host cell due to accumulation of random mutations in the genome[6–8]. In vivo mutagenesis systems with targeted specificity have been reported for *Escherichia coli* and yeast cells, which are the preferred hosts for cloning and expression of gene libraries[4]. For instance, an error-prone variant of *E. coli* DNA polymersase I enables the mutagenesis of genes cloned in ColE1 plasmids, although it concentrates mutations close to the replication origin[9]. A different approach involves the transformation of *E. coli* with mutant oligonucleotide libraries, targeting one or multiple loci in the chromosome, which induce mutation during DNA replication in vivo[10,11]. Iterative cycles of transformation with oligonucleotide libraries followed by high-throughput screenings are needed. In yeast, generation of variability can be achieved by cloning the target DNA segments up to 5 kb in retrotransposons having an error-prone retro-transcriptase[12]. Recently, an in vivo mutagenesis system, named OrthoRep, uses a highly error-prone orthogonal DNA polymerase and a linear plasmid pair that allows the directed evolution of DNA segments of at least 18 kb[13]. However, directed evolution of gene segments or domains is challenging since the orthogonal DNA polymerase replicates the whole plasmid.

A versatile mutagenic system for different hosts is based on the tethering of base deaminases (BDs), such as cytosine deaminases (CDs) and adenosine deaminases (ADs), to a target DNA using catalytically dead Cas9 (dCas9) targeted with a CRISPR RNA (crRNA) or guide RNA (gRNA)[14,15]. Expression of CDs, such as human AID (activation-induced cytidine deaminase), and orthologs from rat (rAPOBEC1, rat apolipoprotein B mRNA editing enzyme 1) and lamprey (pmCDA1, lamprey cytidine deaminase 1), induce random C to T mutations in vivo, both in *E. coli* and yeast. These CDs increase the frequency of C:G to T:A base pair transitions in DNA, especially when uracil DNA N-glycosilase (UNG) activity is inactived by gene deletion or by the specific inhibitor uracil N-glycosylase inhibitor (UGI)[16,17]. Cytosine deamination produces uracil in DNA that can be eliminated by UNG, generating an abasic DNA that is a substrate of the base excision repair system[18]. When a CD is fused to dCas9, its mutagenic activity is tethered to the target DNA sequence hybridized by the crRNA (or gRNA) allowing edition of specific bases in the genome[14,15]. In addition, mutations of A to G (inducing A:T to G:C transitions) have been generated by fusing dCas9 to an engineered AD named TadA* (*E. coli* adenine deaminase variant TadA7.10), derived from the endogenous RNA-dependent AD TadA of *E. coli*[19]. Fusion of these BDs to dCas9 provides precise edition of specific bases in the genome, but its lack of processivity limits its potential for directed evolution. An interesenting alternative is the use of a nickase Cas9 fused to an error-prone DNA polymerase that is able to introduce mutations in a DNA segment of up to 350 bp[20], which is still limited for long genes and operons.

Hence, in vivo mutagenesis systems with high specificity and processivity are of great interest. In this study, we report a strategy that fulfill these criteria enabling in vivo mutagenesis of a target gene (full or partial), using different highly processive protein fusions of BDs and the bacteriophage T7 RNA polymerase (T7RNAP). The specificity of the mutagenesis is provided by a distinct T7RNAP promoter ($P_{T7}$) driving transcription through the target DNA. By placing the $P_{T7}$ at the 3′-end of the target gene, in reverse orientation, expression of the target gene can be preserved from its endogenous 5′-promoter recognized by the host RNAP. Two recent independent studies have also reported the mutagenic activity of CD fusions to T7RNAP, in *E. coli*[21] and in mammalian cells[22]. Moore et al.[21] described a chimeric fusion between rAPOBEC1 and T7RNAP (MutaT7) that in *E. coli* introduces mutations within a plasmid DNA segment downstream of $P_{T7}$. Due to the high processivity of this fusion, an array of at least four copies of the T7 terminator were needed downstream of the target gene to restrict the mutagenesis[21]. The study in mammalian cells uses fusions of rAPOBEC1 or a hyperactive mutant AID to T7RNAP. Mutations in a DNA segment of at least 2000 bp were reported, but no brake system was included to delimit the edited window[22]. In this work, we describe the mutagenic action of different BDs (i.e., AID, rAPOBEC1, pmCDA1, and TadA*) fused to T7RNAP on a target genomic DNA segment in *E. coli* chromosome. Further, we show that a DNA-bound crRNA/dCas9 complex hinders elongation BD-T7RNAP hybrids, protecting the downstream DNA in the chromosome, even within the coding sequence of the target gene, where an arrary of T7 terminators cannot be inserted without disrupting the coding sequence. We also demonstrate that this T7-targeted dCas9-limited in vivo mutagenesis (T7-DIVA) system is useful for continuous directed evolution experiments, using the antibiotic resistant gene *TEM-1*. Given the demonstrated functionality of BDs, T7RNAP, and dCas9 in different hosts[14,15,23,24], T7-DIVA has the potential to be implemented in diverse organisms other than *E. coli*.

## Results

**An *E. coli* reporter strain to test the BD-T7RNAP fusions.** A scheme of the overall strategy followed in this study is shown in Fig. 1a. To measure both the mutagenic and transcriptional activity of BD-T7RNAP fusions, we designed a *gfp-URA3* genetic cassette comprising two gene reporters in reverse orientation: a promoter-less *gfp* gene and the *URA3* gene from *Saccharomyces cerevisiae* (Fig. 1b). Transcription of the *URA3* gene was placed under the Ptac promoter recognized by *E. coli* RNAP. Yeast *URA3* encodes the enzyme orotidine 5′-phosphate decarboxylase involved in the synthesis of uridine monophosphate (UMP)[25]. The activity of *URA3*, and that of the *E. coli* orthologue *pyrF*, allows cell growth in the absence of uracil in the medium (positive selection). In addition, *URA3* expression makes yeast and *E. coli* cells sensitive to 5′-fluororotic acid (FOA) allowing selection of null mutants (negative selection or counterselection)[26,27]. To enable specific recruitment of BD-T7RNAP fusions, the promoter $P_{T7}$ was placed downstream of *URA3*, in reversed orientation to the coding sequence of *URA3*, but in the same orientation that the promoter-less *gfp* gene (Fig. 1b). Thus, expression of GFP acts as a reporter of the transcriptional activity of BD-T7RNAP fusions. The *gfp-URA3* cassette, flanked by transcriptional terminators (T1 and T0), was integrated in the chromosome of *E. coli* K-12 replacing the *flu* gene[28,29]. The *E. coli* K-12 strain used for integration (MG1655*Δ*pyrF*) was derived from the reference strain MG1655[30] having a deletion of *pyrF* and a corrected *pyrE* gene (Supplementary Information) that eliminates a natural mutation reducing its expression[31] (Supplementary Fig. 1a). The *pyrE* gene

is required in the efficient incorporation of FOA to produce the toxic 5-FUMP. Deletion of *pyrF* makes bacteria auxotrophic for uracil and resistant to FOA (Supplementary Fig. 1b). The final reporter strain with the integrated cassette, named MG*-URA3, grows well in mineral media (M9) lacking uracil and is highly sensitive to FOA (Supplementary Fig. 1b). Lastly, an *ung* deletion mutant was obtained from this strain (MG*-URA3Δ*ung*).

**Mutagenic activity of different CD-T7RNAP fusions.** First, we investigated different N- and C-terminal fusions of human CD AID[32] connected with a flexible peptide linker $(G_3S)_7$ to T7RNAP, and some including N-terminal thioredoxin 1 (TrxA) and a cytosolic version of the maltose-binding protein (MBPc). Native T7RNAP and these fusions were cloned under the control of the tetracycline-inducible promoter (TetR-PtetA)[33] in a low copy-number plasmid (pSEVA221)[34] and expressed in MG*-URA3Δ*ung*. These experiments revealed that all N-terminal fusions to T7RNAP produce a transcriptionally active polypeptide in *E. coli* with similar mutagenic activity (Supplementary Notes, and Supplementary Figs. 2 and 3a). We chose the AID-T7RNAP fusion, lacking any additional protein partner, to continue our work. Then, we constructed similar N-terminal fusions with other CDs, namely pmCDA1 and rAPOBEC1, in the same vector (Fig. 2a). MG*-URA3Δ*ung* strains carrying plasmids encoding native T7RNAP, AID-T7RNAP, pmCDA1-T7RNAP,

rAPOBEC1-T7RNAP, or pSEVA221 (negative control), were grown and induced with anhydrotetracycline (aTc). Western blot analysis of whole-cell protein extracts revealed slighlty higher expression levels of pmCDA1-T7RNAP and rAPOBEC1-T7RNAP than AID-T7RNAP, and the overexpression of native T7RNAP (Fig. 2b). Flow cytometry analysis showed similar levels of GFP in bacteria encoding AID, pmCDA1, and rAPOBEC1 fusions, roughly half of those found in bacteria with T7RNAP (Fig. 2c and Supplementary Fig. 3b). Thus, all CD-T7RNAP fusions were expressed and transcriptionally active in *E. coli*.

Next, we determined the mutant frequency in *URA3* (on-target) and *rpoB* (off-target) upon induction of native T7RNAP, AID, pmCDA1, rAPOBEC1 fusions, and pSEVA221, in three different *E. coli* strains: MG*-URA3 (*ung*⁺), MG*-URA3Δ*ung*, and MG*-URA3Δ*ung*ΔP_T7 (lacking the P_T7 in *URA3*). Bacteria were plated on M9 + uracil and M9 + uracil + FOA to determine the *URA3* mutant frequency for each strain as the ratio of FOA^R colony forming units (CFU ml⁻¹) vs. total CFU ml⁻¹ (Fig. 2d). The frequency of *URA3* mutants in MG*-URA3Δ*ung* bacteria was ~$10^{-6}$ for the control (pSEVA221), ~$10^{-5}$ for native T7RNAP, and increased to ~$10^{-3}$ for AID-T7RNAP, to ~$5 \times 10^{-2}$ for rAPOBEC1-T7RNAP, and up to ~$10^{-1}$ for pmCDA1-T7RNAP (Fig. 2d). Thus, all CD fusions have a clear mutagenic activity over control strain (≥1000-fold) in the Δ*ung* mutant, with rAPOBEC1 and pmCDA1 fusions having a higher activity than AID fusion (~50- to 100-fold, respectively). All CD fusions showed a much lower mutagenic activity in the *ung*⁺ strain (<1% that in Δ*ung*; Fig. 2d). Importantly, the frequency of *URA3* mutants for all CD fusions and native T7RNAP dropped to background levels in the strain lacking the T7 promoter (MG*-URA3Δ*ung*ΔP_T7 strain; Fig. 2d). Noticiable, induction of native T7RNAP increased *URA3* mutation rate ~10-fold in a P_T7-dependent manner, but irrespective of UNG (Fig. 2d). Bacteria from these cultures were also plated on rifampicin (Rif)-containing plates to evaluate the specificity of the fusions. Rif^R colonies of *E. coli* are known to contain mutations in *rpoB*, encoding the β-subunit of *E. coli* RNAP[35]. Thus, Rif^R was used to determine the off-target *rpoB* mutant frequency as the ratio of Rif^R CFU ml⁻¹ vs. total CFU ml⁻¹ (Fig. 2e). The frequency of spontaneous *rpoB* mutants of the control strain (~$10^{-7}$) was only mildly increased (~2 to 5-fold) by the expression of AID fusion, and ~10–20-fold with the more active pmCDA1 and rAPOBEC1 fusions (Fig. 2e). Hence, the mutagenic activity of CD-T7RNAP fusions in *URA3* requires the presence of P_T7, is more efficient in the Δ*ung* mutant and has a strong preference for the target *URA3* gene vs. off-target *rpoB*.

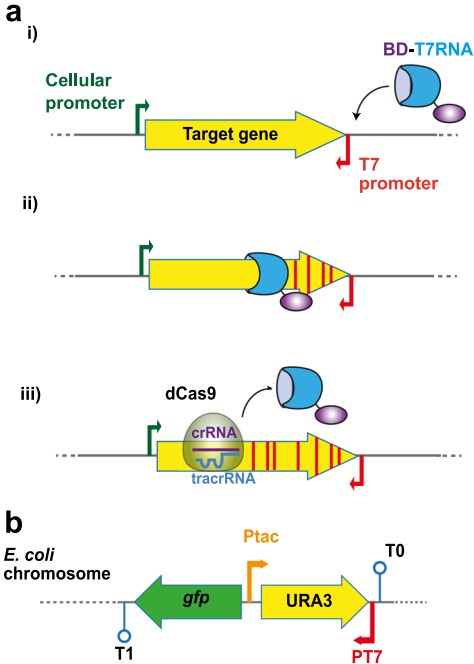

**Fig. 1 Graphic summary of the mutagenesis system development.**
**a** Schematic representation of the mutagenic process. The T7 RNA polymerase fusion (BD-T7RNAP, blue shape joined by a black line to a purple elliptical shape) specifically binds the T7 promoter (i), initiating the transcription and moving along the target gene (yellow filled arrow) carrying the base deaminase (BD) that introduces mutations (red stripes) in the gene (ii). The fusion stops and detaches from the DNA when encounters a dCas9 molecule (translucid ovoid shape) bound to a specific sequence determined by the CRISPR RNA (crRNA, purple line). The *trans*-activating CRISPR RNA (tracrRNA, blue line) is also required for this process (iii). **b** Representation of the chromosomally integrated reporter *gfp-URA3* cassette to test the mutagenesis system. The genes *gfp* and *URA3* are represented with green and yellow filled arrows, respectively. Thin arrows indicate the promoters tac (P_tac) and T7 (P_T7), lollipops indicate terminators T0 and T1.

**Mutagenic activity of TadA*-T7RNAP fusion.** To broaden the mutagenic capacity of the system, a fusion was constructed with TadA*[19] (Fig. 3a). TadA* deaminate adenines in DNA generating inosines that lead to A:T > G:C transitions. TadA*-T7RNAP was expressed in MG*-URA3Δ*ung* at higher levels than AID-T7RNAP, as determined by western blot (Fig. 3b), but produced only slightly higher level of GFP (Fig. 3c and Supplementary Fig. 3c). The mutagenic capacity of TadA*-T7RNAP was evaluated in different genetic backgrounds, using AID-T7RNAP and pSEVA221 as positive and negative controls, respectively (Fig. 3d). Expression of TadA*-T7RNAP generated *URA3* mutants with a frequency of ~$2–5 \times 10^{-4}$ (~100-fold that of the control) in both *ung*⁺ and Δ*ung* strains, indicating that TadA* activity is independent of UNG (Fig. 3d). The gene *nfi* encodes the endonuclease V of *E. coli*, which eliminates inosines[36,37]. When TadA*-T7RNAP was expressed in Δ*nfi* mutants (MG*-URA3Δ*nfi* and MG*-URA3Δ*ung*Δ*nfi*), the frequency of *URA3* mutants increased to ~$10^{-3}$, similar to that of AID-T7RNAP in Δ*ung* (Fig. 3d). Deletion of *nfi* had no effect on AID-T7RNAP (Fig. 3d). Notably,

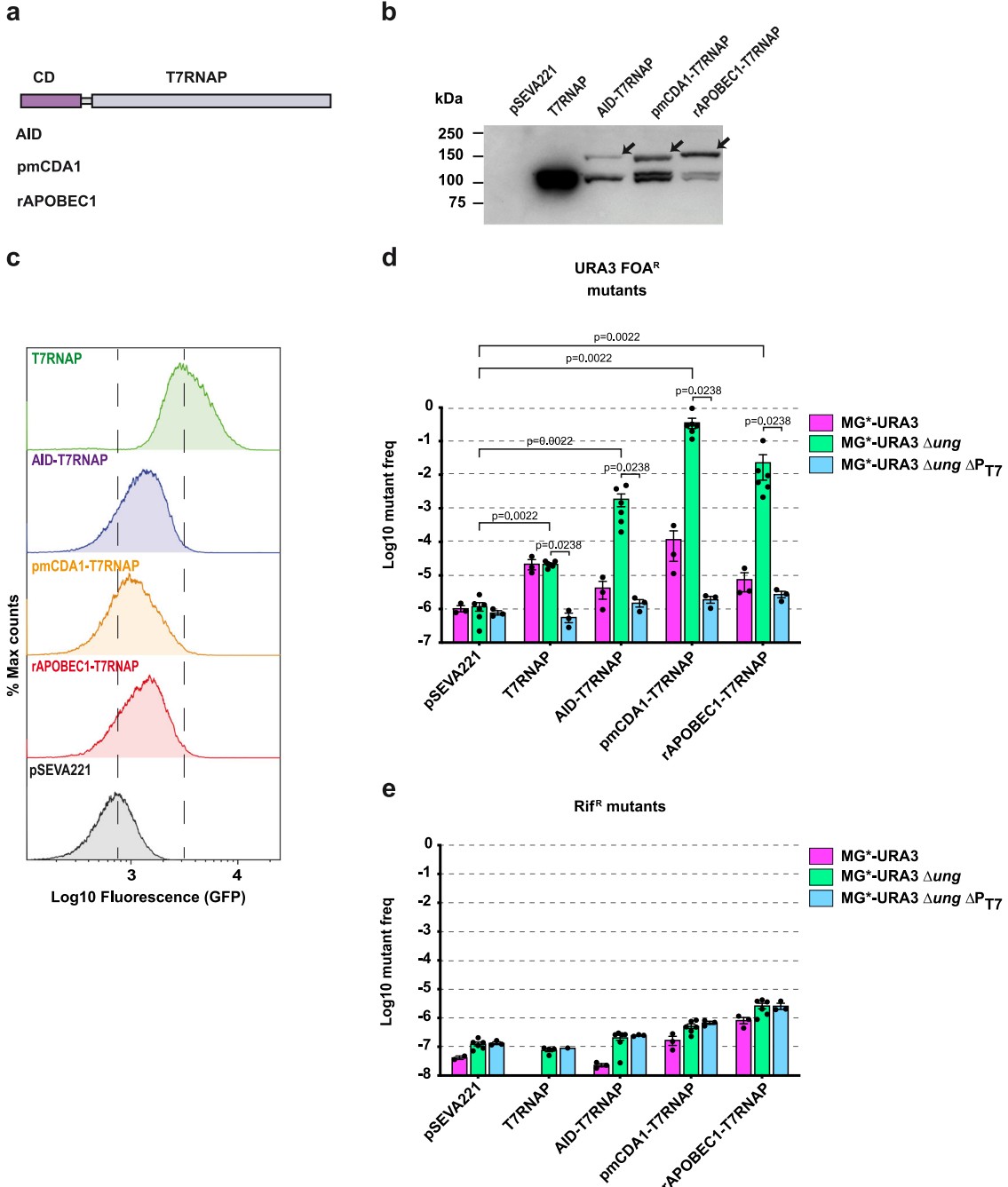

**Fig. 2 Expression and mutagenic activity of AID-T7RNAP, pmCDA1-T7RNAP, and rAPOBEC1-T7RNAP. a** Scheme of the different CDs fused to the T7RNAP by the linker $(G_3S)_7$. **b** Expression of the different fusions determined by western blot analysis of the cell extracts from induced cultures of the strain MG*-URA3Δ*ung*. A representative immunoblot is shown from two independent experiments with similar results. **c** Processivity of the fusions assessed by flow cytometry analysis to detect expression of GFP in the induced cultures. The gating strategy and the corresponding pseudocolor plots are shown in Supplementary Fig. 3b. **d**, **e** Mutagenic activity in *URA3* (**d**) and *rpoB* (**e**) of the different fusions using as hosts MG*-URA3, MG*-URA3Δ*ung*, and MG*-URA3Δ*ung*ΔP$_{T7}$. The histograms **d** and **e** show the single values (black dots), the means (bars), and standard errors (lines) of at least three independent experiments ($n = 3$, except for the strain MG*-URA3Δ*ung* $n = 6$). The statistical analysis was done using two-tailed Mann–Whitney test. Exact *p* values (*p*) are indicated in the figure. A *p* value $< 0.05$ was considered significant. Source data are provided as a Source data file.

expression of TadA*-T7RNAP did not produce any significant increase in the levels of off-target mutagenesis in *rpoB* (Fig. 3e). In addition, the mutagenic activity of TadA*-T7RNAP in *URA3* requires the presence of P$_{T7}$, dropping to background levels in MG*-URA3Δ*ung*ΔP$_{T7}$ (Fig. 3f). These data demonstrate that TadA*-T7RNAP fusion has a specific mutagenic activity for the target DNA having P$_{T7}$. This activity is independent of UNG and increases moderately when endonuclease V is absent.

**Characterization of the mutations.** We randomly picked 30 FOA[R] colonies (*URA3* mutants) from each of the MG*-URA3Δ*ung* strains expressing native T7RNAP, AID, pmCDA1, and rAPOBEC1 fusions, and 30 additional FOA[R] colonies from MG*-URA3Δ*ung*Δ*nfi* expressing TadA*-T7RNAP. The chromosomal Ptac-*URA3*-P$_{T7}$ region from these colonies was amplified by PCR and sequenced. As a control, identical region was sequenced from 13 FOA-sensitive (nonmutant) colonies from

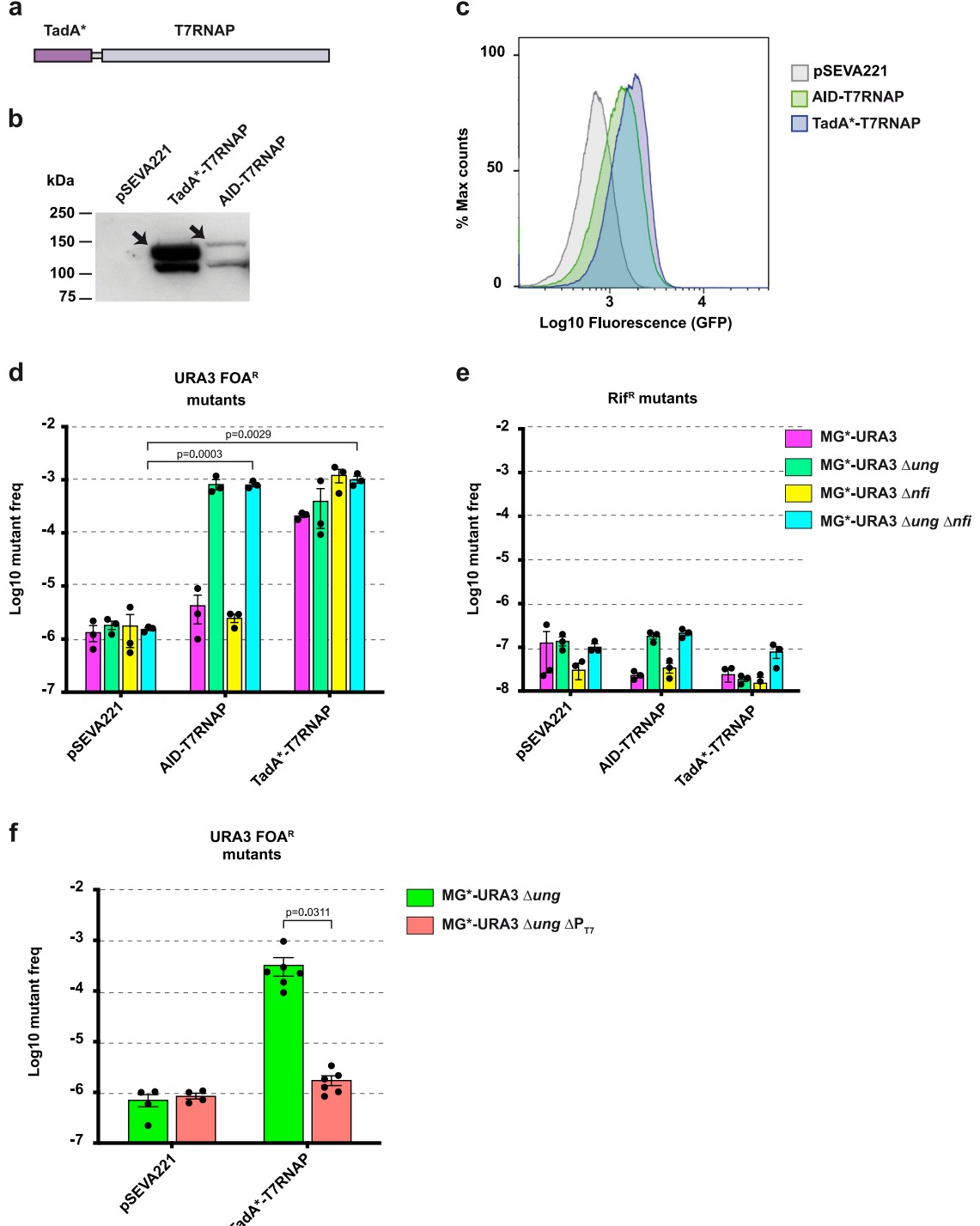

**Fig. 3 Expression and activity of TadA*-T7RNAP. a** Scheme of the fusion TadA*-T7RNAP with the linker $(G_3S)_7$. **b** Expression of the fusion TadA*-T7RNAP in comparison to AID-T7RNAP determined by western blot analysis of cell extracts from induced cultures. A representative immunoblot is shown from two independent experiments with similar results. **c** Processivity of the fusions assessed by flow cytometry analysis to detect expression of GFP in the induced cultures. The gating strategy and the corresponding pseudocolor plots are shown in Supplementary Fig. 3c. **d, e** Mutagenic activity of the AID- and TadA*-T7RNAP fusions in *URA3* (**d**) and *rpoB* (**e**), using as hosts MG*-URA3, MG*-URA3Δ*ung*, MG*-URA3Δ*nfi*, and MG*-URA3Δ*ung*Δ*nfi* ($n = 3$ independent experiments). **f** *URA3* mutant frequency when TadA*-T7RNAP is expressed in MG*-URA3Δ*ung* and MG*-URA3Δ*ung*ΔP$_{T7}$ (pSEVA221 $n = 4$, TadA*-T7RNAP $n = 6$). The histograms (**d–f**) show the single values (black dots), the means (bars), and standard errors (lines) of multiple independent experiments. The statistical analysis was done using two-tailed Student's *t* test. Exact *p* values (*p*) are indicated in the figure. A *p* value < 0.05 was considered significant. Source data are provided as a Source data file.

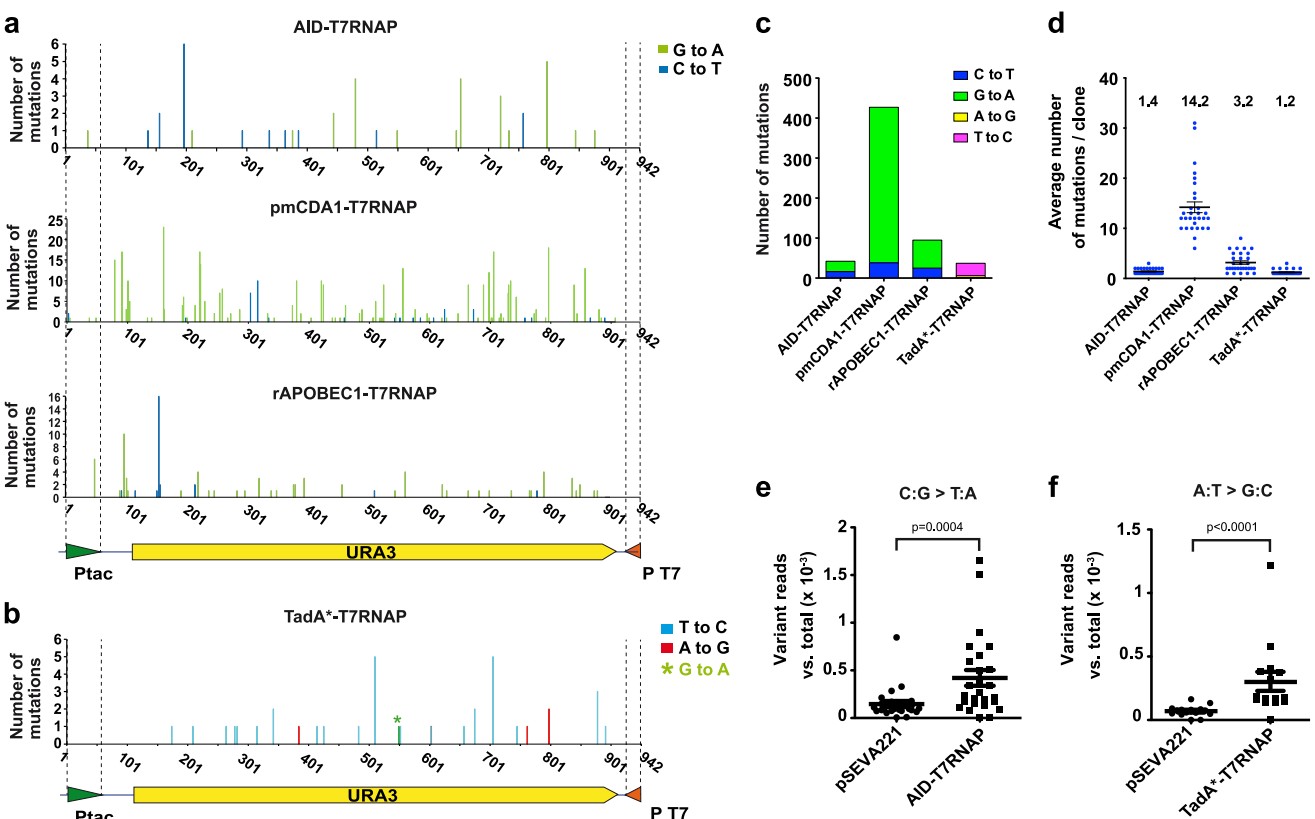

**Fig. 4 Characterization of *URA3* mutations found in FOA^R colonies expressing BD-T7RNAP fusions. a** Number of mutations per nucleotide identified in the *URA3* locus from 30 FOA^R colonies isolated from each MG*-URA3Δ*ung* strain expressing the indicated CD-T7RNAP fusions and **b** from MG*-URA3Δ*ung*Δ*nfi* strain expressing TadA*-T7RNAP fusion. The promoters Ptac and T7 are shown with green and red arrow heads, respectively, and delimited by dashed lines. The gene *URA3* is shown with a yellow filled arrow. The indicated base changes correspond to the coding sequence of *URA3*. Different base substitutions found are labeled with the color codes on the right. A single G to A transition found with TadA*-T7RNAP is labeled with an asterisk. **c** Total number of mutations for each BD-T7RNAP fusion indicating the base substitutions found. **d** Average number of mutations per clone found in the FOA^R colonies analyzed for each of the indicated BD-T7RNAP fusions. Single values are represented with blue dots and means and standard errors with black lines. **e**, **f** Variant calling analysis of a 200 bp region of *URA3* after its massive DNA sequencing (ca. 10^6 reads). The number of reads with different variants vs. total reads are represented with circles (empty plasmid) and squares (AID-T7RNAP or TadA*-T7RNAP). The lines represent the means and the standard errors from each group. The statistical analysis was done using two-tailed Mann–Whitney test. Exact *p* values (*p*) are indicated in the figure. A *p* value < 0.05 was considered significant. Source data are provided as a Source data file.

MG*-URA3Δ*ung* (pSEVA221), showing no mutation from wild-type allele. In contrast, all FOA^R colonies expressing CD-T7RNAP fusions had multiple transitions C:G > T:A in both DNA strands along the Ptac and *URA3* gene, but none in P$_{T7}$ (Fig. 4a). *URA3* alleles from bacteria expressing TadA*-T7RNAP contained transitions A:T > G:C in both DNA strands, except a single C:G > T:A transition (Fig. 4b). No other type mutations, deletions, or insertions, were observed in any *URA3* alleles from BD-T7RNAP fusions. This was not the case in FOA^R colonies expressing native T7RNAP, which contained different types of mutations, including transitions, transversions, deletions, and insertions in *URA3* (Supplementary Fig. 4). Showing correlation to the mutagenic capacity of the BD-T7RNAP fusions, the highest total number of mutations was found with pmCDA1-T7RNAP (426) followed by rAPOBEC1-T7RNAP (95), AID-T7RNAP (42) and TadA*-T7RNAP (37) (Fig. 4c). The average number of mutations per clone presented the same hierarchy: pmCDA1-T7RNAP (14.2) > rAPOBEC1-T7RNAP (3.2) > AID-T7RNAP (1.4) > TadA*-T7RNAP (1.2) (Fig. 4d). For all CD-T7RNAP fusions, transitions G to A were detected more frequently than C to T in the *URA3* coding strand (Fig. 4a, c), indicating a higher mutation rate of Cs in the noncoding strand of *URA3*, which corresponds to the non-template strand for the CD-T7RNAP fusions (Supplementary Fig. 5). This bias toward the

non-template strand of T7RNAP is less pronounced for AID (62%) than for rAPOBEC1 (74%) or pmCDA1 (91%) fusions (Fig. 4c). For TadA* fusion, we also found a bias favoring T to C mutations in the coding strand of *URA3* (84%), corresponding to A to G mutations in the non-template strand of T7RNAP (Fig. 4c and Supplementary Fig. 5). Therefore, DNA sequencing of *URA3* mutants demonstrates that BD-T7RNAP fusions induce the expected mutations with a bias toward the non-template strand.

For deeper analysis, a 284 bp PCR fragment of the *URA3* gene was amplified from an induced culture of MG*-URA3Δ*ung* expressing AID-T7RNAP without FOA selection, and was subjected to massive next-generation sequencing (NGS). The same region was subjected to NGS from a control MG*-URA3Δ*ung* (pSEVA221). Comparison of the variant call analysis from the two samples (ca. $1 \times 10^6$ reads/sample) indicated that only the transition C:G > T:A appeared in an statistically significant higher number in bacteria expressing AID-T7RNAP (Fig. 4e and Supplementary Fig. 6). We used the same approach to analyze the induced *URA3* mutations by TadA*-T7RNAP in MG*-URA3Δ*ung*Δ*nfi* compared with the same strain carrying pSEVA221. In this case, only transitions A:T > G:C were detected in a statistically significant higher number in the sample expressing TadA*-T7RNAP (Fig. 4f and Supplementary Fig. 7). Hence, massive DNA sequencing data is consistent with the DNA

sequencing results of individual *URA3* mutants, and confirms that BD-T7RNAP fusions generate only the expected mutations.

We also confirmed the adequacy of the Rif[R] phenotype as reporter of the off-target activity of CD and AD fusions. The Rif[R] phenotype is mostly caused by *rpoB* mutations in a region between amino acid 507 and 687 of the β-subunit of *E. coli* RNAP known as RIF-resistant determing region (RRDR)[38]. We sequenced the *rpoB*-RRDR segment in 30 Rif[R] colonies from strains MG*-URA3Δ*ung* and MG*-URA3Δ*nfi* carrying pSEVA221. Among other mutations, transitions C to T and G to A were readily found in *rpoB*-RRDR from Rif[R] MG*-URA3Δ*ung* (Supplementary Fig. 8a). Similarly, Rif[R] MG*-URA3Δ*nfi* contained diverse transitions A to G and T to C in *rpoB*-RRDR (Supplementary Fig. 8b). Consequently, different bases within *rpoB*-RRDR that are sensitive to CD and AD mutagenesis can generate Rif[R] mutants, validating *rpoB* to assess the off-target activity of BDs.

**Protection of downstream regions using dCas9.** The BD-T7RNAP fusions were able to transcribe the *gfp* gene downstream of *URA3* (respective to the elongation of T7RNAP) potentially generating mutations beyond *URA3*. To confirm this, we inserted *sacB* from *Bacillus subtilis* in the *gfp-URA3* cassette as an additional counterselection gene[39]. *B. subtilis* *sacB* codes for the exoenzyme levansucrase, which utilizes sucrose to produce toxic levan, which accumulates in the periplasm of *E. coli* killing the bacterium. A Ptac-*sacB* fragment was cloned in *gfp-URA3* and the new *sacB-gfp-URA3* cassette (Fig. 5a) was integrated replacing the chromosomal *flu* gene in MG*Δ*pyrF*Δ*ung*Δ*nfi*. The resulting strain, MG*-SacB-URA3Δ*ung*Δ*nfi*, was sensitive to sucrose with a frequency of spontaneous mutants of $\sim 5 \times 10^{-6}$. When AID-T7RNAP was expressed in this strain, the frequency of *sacB* mutants increased to $\sim 6 \times 10^{-4}$ (Supplementary Fig. 9), whereas that of *URA3* ($\sim 1.7 \times 10^{-3}$) was similar to that observed previously (Fig. 2d). This confirms that BD-T7RNAP fusions are able to mutate regions downstream of the target gene.

To delimit the mutagenic activity within the target gene, we investigated the possibility of blocking elongation of BD-T7RNAP with dCas9. dCas9 can bind a target DNA sequence using crRNAs (or gRNAs) and has been used as transcriptional repressor for the endogenous *E. coli* RNAP[40]. Co-expression of two gRNAs targeting the non-template strand of a gene enhances the transcription repression of *E. coli* RNAP by dCas9[41]. Therefore, we designed two crRNAs (a and b) against the non-template strand of *gfp* (Fig. 5a, b), generating a double crRNA array (b.a). Plasmid pdCas9[40] was used for the constitutive expression of dCas9, the *trans*-activating CRISPR RNA (tracrRNA), and the designed crRNA arrays. MG*-SacB-URA3Δ*ung*Δ*nfi* strains carrying pdCas9b.a, or control pdCas9, were transformed with the plasmid encoding AID-T7RNAP or pSEVA221 control. After growth and aTc induction, we determined that GFP expression by AID-T7RNAP was reduced by the double crRNAb.a to ~30% of that produced in the isogenic strain lacking crRNA (Fig. 5c). Basal levels of GFP were detected in strains carrying pSEVA221 and were considered 0. Protection of *sacB* from AID-T7RNAP mutagenesis was assessed by the ratio of the mutation in *sacB* vs. *URA3* for each strain. This ratio was normalized as 1 for the strain expressing AID-T7RNAP and carrying pdCas9 without crRNAs (Fig. 5d). In concordance with the reduction of GFP expression, the mutagenesis of *sacB* dropped ~10-fold when the double crRNAb.a was expressed (Fig. 5d). Then, we tested whether the blockade of AID-T7RNAP could be enhanced with a triple (b.a.c) crRNA array (Fig. 5a). Expression of dCas9 and crRNAb.a.c repressed the levels of GFP to ~20% of those found with pdCas9 lacking crRNA, and the

mutant frequency in *sacB* vs. *URA3* decreased ~14-fold (Fig. 5c, d). Therefore, dCas9 directed with crRNAs is able to hamper the progression and mutagenesis of BD-T7RNAP fusions along the DNA, being a triple crRNA array more effective.

To evaluate whether dCas9 can be used to protect a particular region within a gene, we designed a new triple crRNA array (d.e.f) targeting *URA3* (Fig. 6a). Cultures of MG*-SacB-URA3Δ*ung*Δ*nfi* strain expressing AID-T7RNAP and dCas9 with crRNAd.e.f (targeting *URA3*), or crRNAb.a.c (targeting *gfp*) as a control, were induced and the *URA3* alleles from 30 FOA[R] colonies from each culture were sequenced. As expected, mutations in FOA[R] clones expressing crRNAb.a.c (targeting *gfp*) were found distributed all along *URA3*, whereas mutations in FOA[R] clones expressing crRNAd.e.f (targeting *URA3*) were only found in the gene segment between the recognition sites of the crRNAs and P$_{T7}$ (Fig. 6b). Interestingly, albeit mutations in *URA3* were not detected more distal than the hybridization site of crRNA d, we found mutations in regions between the hybridization sites of crRNAs d to e and e to f (Fig. 6b), as well as in the *URA3* segment proximal to P$_{T7}$. Collectively, these results demonstrate that targeted dCas9 blockade with a triple crRNA array can be used to concentrate the mutagenesis activity of BD-T7RNAP fusions to a specific target gene or gene segment, reducing the mutagenesis of downstream DNA regions.

**Fast directed evolution of the TEM-1 gene.** To test this in vivo mutagenesis system in a directed evolution process, we chose the antibiotic resistance gene *TEM-1* as proof-of-principle. This gene encodes the TEM-1 β-lactamase that confers resistance to penicillins, cephalosporins, and related β-lactams[42]. The evolution of this enzyme has been extensively documented due to its clinical relevance, with the description of >170 variants, some of them having increased resistance to third-generation cephalosporins, such as ceftazidime (CAZ)[42]. We planned to evolve TEM-1 in vivo to obtain variants that confer resistance to CAZ. To this end, the *URA3* gene in the *sacB-gfp-URA3* cassette was replaced by *TEM-1*, and the new *sacB-gfp-TEM-1* cassette was inserted, replacing *flu*, in the chromosome of MG*Δ*pyrF*Δ*ung*Δ*nfi* (Fig. 7a). The resulting strain (MG*-SacB-TEM-1*ung*Δ*nfi*) was transformed with plasmids encoding AID-T7RNAP and dCas9-crRNAb.a.c (to protect downstream genes). Three independent transformants were cultured separately and subjected to two iterative cycles of growth and induction with aTc (Fig. 7b). As controls of non-induced evolution (i.e., spontaneous *TEM-1* mutation), three colonies from the same strain carrying the empty vectors (pSEVA221 and pdCas9) were grown and induced in parallel. At the end of each induction cycle, serial dilutions of each culture were plated on LB agar with increasing concetrations of CAZ (0, 1, 4, and 16 μg ml$^{-1}$) to calculate the frequency of CAZ[R]-mutants at each concentration, expressed as the ratio of CAZ[R] CFU ml$^{-1}$ vs. total CFU ml$^{-1}$ (Fig. 7c). After one cycle, the frequency of CAZ[R]-mutants in cultures expressing AID-T7RNAP and dCas9-crRNAb.a.c was $\sim 6.5 \times 10^{-5}$ at 1 μg ml$^{-1}$, and $\sim 3 \times 10^{-7}$ at 4 μg ml$^{-1}$ of CAZ, with no resistant colonies detected to 16 μg ml$^{-1}$. In contrast, spontaneous CAZ[R]-mutants in control cultures appeared at ~2000-fold lower frequency at 1 μg ml$^{-1}$ of CAZ, and no resistant colonies appeared at any higher concentration (Fig. 7c). Interestingly, bacterial cultures expressing AID-T7RNAP, and subjected to two cycles of growth and induction increased the frequency of CAZ[R]-mutants at 1 and 4 μg ml$^{-1}$ of CAZ by ~10-fold ($\sim 5 \times 10^{-4}$ and $\sim 2 \times 10^{-6}$, respectively) and CAZ[R] colonies arose at the highest CAZ concentration (16 μg ml$^{-1}$) with a frequency of $\sim 2 \times 10^{-7}$ (Fig. 7c). In contrast, after two cycles of growth, spontaneous CAZ[R]-mutants were not detected in control cultures at 4 or 16 μg ml$^{-1}$ of CAZ, and CAZ[R] clones only

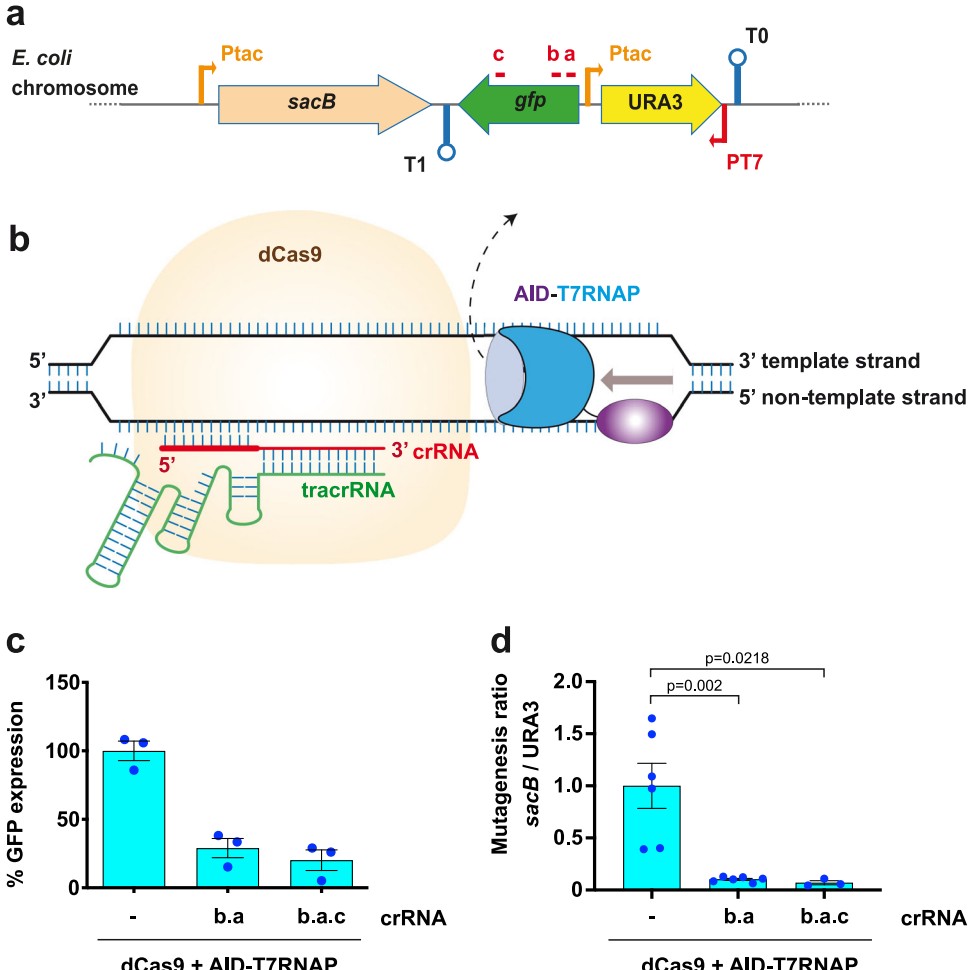

**Fig. 5 Blocking AID-T7RNAP elongation and mutagenic activity by dCas9 and crRNAs. a** Scheme of the reporter cassette with *sacB* integrated in MG*-SacB-URA3Δ*ung*Δ*nfi* strain. Thin arrows indicate the tac ($P_{tac}$) and T7 ($P_{T7}$) promoters, lollipops indicate terminators T0 and T1; red lines mark targeting sequences of the crRNAs a, b, and c; filled arrows indicate the genes *sacB* (orange), *gfp* (green), and *URA3* (yellow). **b** Representation of the dCas9 blocking activity showing one crRNA (in red) targeting the non-template strand relative to T7RNAP transcription. The direction of the progression of the fusion along the DNA is indicated with a gray arrow. The mutagenic protein fusion (AID-T7RNAP) is displaced from the transcription bubble (dashed arrow) by bound dCas9/crRNA. The orange shape represents the dCas9, the blue shape joined by a black line to a purple elliptical shape represents the fusion AID-T7RNAP, the red RNA molecule represents the crRNA and the green RNA molecule represents the tracrRNA. **c** Relative GFP levels measured by flow cytometry of bacteria from strain MG*-SacB-URA3Δ*ung*Δ*nfi* expressing AID-T7RNAP and dCas9 in the absence (−) or presence of crRNA arrays b.a and b.a.c. The histogram shows the percentages of mean fluorescence intensities (MFI) for each condition relative to the strain lacking crRNAs. Background GFP fluorescence signals from this strain with pdCas9 and the empty vector pSEVA221 are subtracted from all values ($n = 3$ independent experiments). **d** Ratio of mutagenesis of *sacB* vs. *URA3* in bacteria MG*-SacB-URA3Δ*ung*Δ*nfi* expressing AID-T7RNAP and dCas9 in the absence (−) or presence of crRNA arrays b.a and b.a.c. The ratio found in bacteria lacking crRNAs are considered 1 (control $n = 6$, b.a $n = 6$, b.a.c $n = 3$). For **c** and **d**, the histograms represent the single values (blue dots), the relative means (bars), and standard errors (black lines) from multiple independent experiments. The statistical analysis was done using two-tailed Student *t* test. Exact *p* values (*p*) are indicated in the figure. A *p* value < 0.05 was considered significant. Source data are provided as a Source data file.

appeared at low frequency ($\sim 1 \times 10^{-7}$) in plates containing $1\,\mu g\,ml^{-1}$ of CAZ. These data indicate that AID-T7RNAP was producing mutations within *TEM-1* at a significant rate above spontaneous mutation (>1000-fold/mutagenic cycle), with mutations being accumulated in each cycle generating variants with different levels of resistance to CAZ.

To characterize the *TEM-1* variants from the evolved cultures expressing AID-T7RNAP, we sequenced *TEM-1* alleles from 12 CAZ$^R$ colonies (four from each triplicate culture) were isolated at $1\,\mu g\,ml^{-1}$ CAZ after cycle 1, and from 12 CAZ$^R$ colonies isolated at $16\,\mu g\,ml^{-1}$ CAZ after cycle 2. This revealed that all mutations found in the *TEM-1* alleles were the expected from the CD activity of AID, being transitions G > A more frequent than C > T

in the coding strand of *TEM-1* (Table 1). All sequenced *TEM-1* variants from $1\,\mu g\,ml^{-1}$ CAZ had mutations in residue R164 (R164C or R164H), and only two clones contained additional mutations (A249V and G267R). In contrast, all *TEM-1* variants selected at $16\,\mu g\,ml^{-1}$ CAZ after two cycles invariable contained R164H and E104K mutations, frequently associated to additional mutations (e.g., V33I and A150T). Mutations R164H and E104K are reported to provide increased resistance to CAZ[42]. We confirmed this determining the minimal inhibitory concentration (MIC) to CAZ of mutant R1.10 (R164H)[43] and of mutant R2.2 (E104K and R164H)[44] (Table 2), demonstrating that *TEM-1* variants with increased resistance to CAZ were obtained in a fast and continuous manner using this in vivo mutagenesis system.

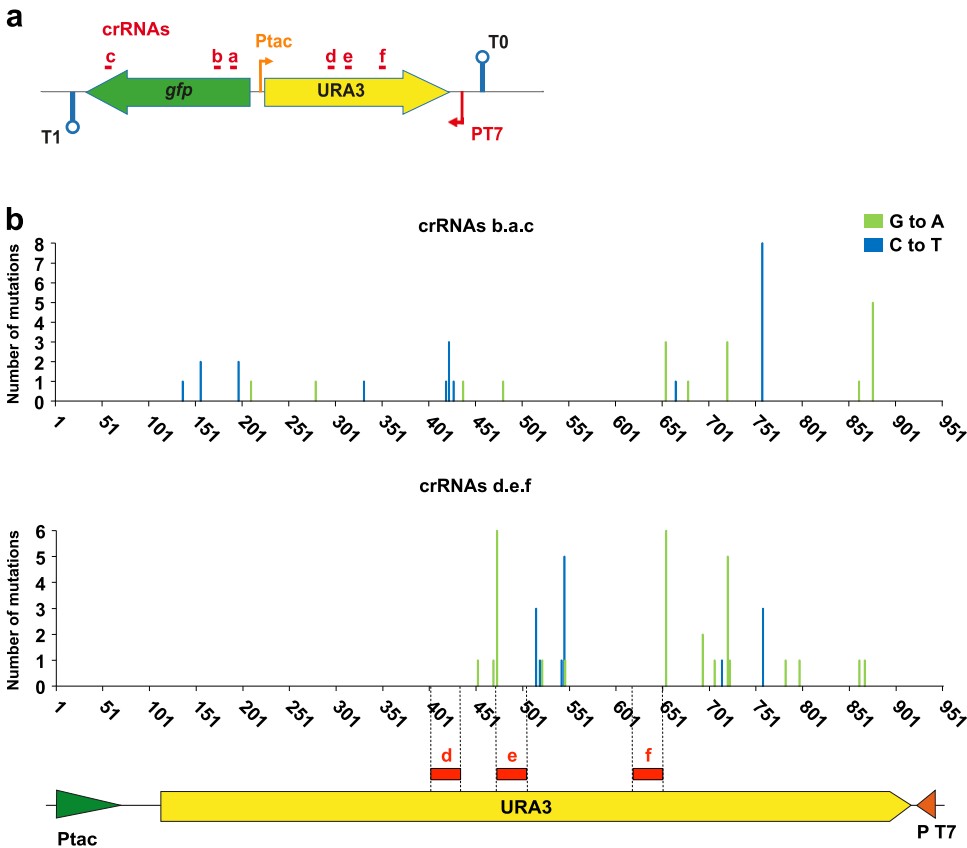

**Fig. 6 Protection from mutagenesis of a delimited segment of the target gene by dCas9 guided with crRNAs. a** Scheme of the *URA3* cassette with the recognition sites (red lines) of the triple crRNAs b.a.c and d.e.f. Thin arrows indicate the tac ($P_{tac}$) and T7 ($P_{T7}$) promoters, lollipop shapes indicate terminators T0 and T1. The genes *gfp* and *URA3* are represented with green and yellow filled arrows, respectively. **b** *URA3* mutations found in FOA$^R$ colonies of induced cultures of MG*-SacB-URA3Δ*ung*Δ*nfi* expressing AID-T7RNAP and the corresponding tripe crRNAs. For each culture the *URA3* alleles from 30 colonies were sequenced. The indicated base changes correspond to the coding sequence of *URA3*. Different base substitutions found are labeled with the color code on the right. The promoters Ptac and T7 are shown with green and red arrow heads, respectively. The gene *URA3* is shown with a yellow filled arrow. Red rectangles over the *URA3* gene mark targeting sequences of the crRNAs d, e, and f, and delimited with dashed lines. Source data are provided as a Source data file.

## Discussion

The in vivo mutagenesis system reported in this work is highly specific for the target gene based on the tethering of BDs fused to T7RNAP, which selectively recognizes $P_{T7}$. By placing $P_{T7}$ in reverse orientation at the 3′-end of the target gene, expression from its endogenous 5′-end promoter can be maintained. Expression of the BD-T7RNAP fusions is inducible with aTc. An inducible expression of the mutagenenic enzyme reduces potential toxicities associated to the constitutive expression of BD and/or T7RNAP, and it is valuable to control the mutagenic process. The use of different BDs (AID, rAPOBEC1, pmCDA1, and TadA*) confers flexibility to the system due to their different mutagenesis profile and activities. For CD-T7RNAP fusions, AID was the less active and pmCDA1 was the most active. TadA*-T7RNAP broads the mutation spectrum generating transitions A:T > G:C with a mutant frequency similar in Δ*nfi* strains to AID-T7RNAP in Δ*ung* strains (ca. $10^{-3}$). A single mutant isolated from TadA*-T7RNAP contained an unexpected C:G > T:A, which could be caused by a nonspecific activity of TadA* or by the use of a Δ*ung* background. Massive DNA sequencing of *URA3* without selection detected only the expected transitions C:G > T:A for AID and A:T > G:C for TadA*. Importantly, no other type of mutations (e.g., deletions and insertions) were found among the mutants isolated after expression of the BD-T7RNAP fusions. This contrasts with the various types of *URA3* mutations found after expression of native T7RNAP. Transcription by native

T7RNAP increases the mutant frequency by a mechanism that is independent of BDs and UNG, but which might be related to conflicts of high transcription with other cellular machineries (e.g., replisome)[45].

Elimination of the DNA repair enzymes UNG (Δ*ung*) and endonuclease V (Δ*nfi*) enabled to obtain higher mutant frequencies for BDs. An interesting alternative to *ung* deletion for CDs is its transient expression of UGI from bacteriophage PBS2[17]. TadA* fusion was not affected by UNG, and its activity was increased moderately by the lack of endonuclease V. This might be due to a lower activity of endonuclease V for the removal of inosines generated by TadA*[19]. Lack of endonuclease V has no effect on the activity of AID fusion.

Also interesting is that the mutagenic activity of all BD-T7RNAP fusions is biased toward the non-template strand. This phenomenon is less pronounced for AID fusion, but it is found in all of them. This observation was also reported for rAPOBEC1 in MutaT7[21]. The reason of this strand preference is unclear, but it might be caused by a higher exposure of bases in the non-template strand at the transcription bubble. The template strand is less accesible due its insertion in the catalytic core of the T7RNAP and the formation of the ssDNA:RNA hybrid[46].

A key property of a targeted mutagenesis system is the specificity toward the on-target sequence, keeping as low as possible the off-target mutagenesis. The specifity of T7-DIVA system was assesed by two different approaches: monitoring mutations in

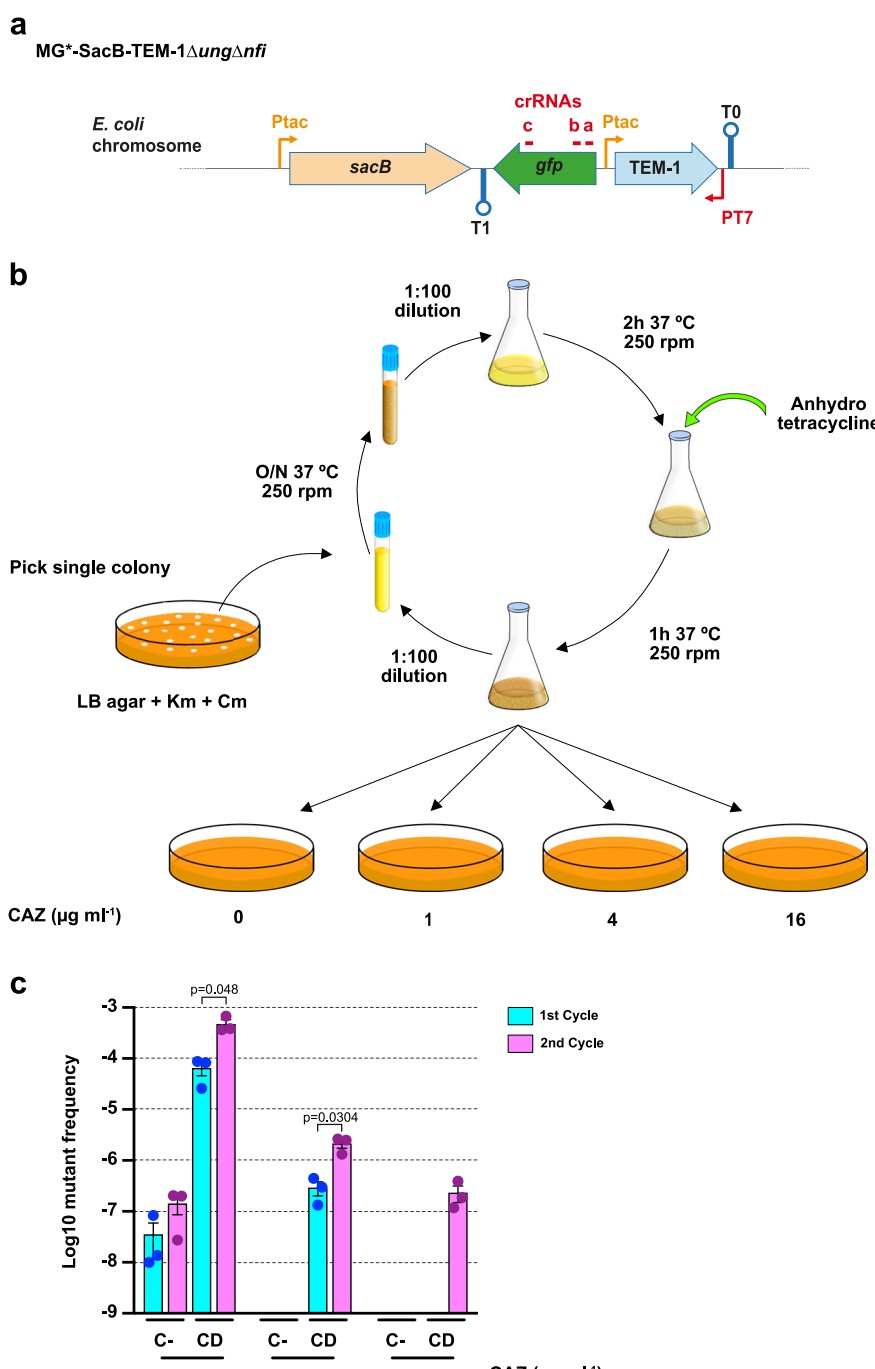

**Fig. 7 TEM-1 evolution using the mutagenesis system. a** Scheme of the cassette with the *TEM-1* gene in the strain MG\*-SacB-TEM-1Δ*ung*Δ*nfi*. Thin arrows indicate the tac (P$_{tac}$) and T7 (P$_{T7}$) promoters, lollipop shapes indicate terminators T0 and T1. The genes *sacB*, *gfp*, and *URA3* are represented with orange, green, and yellow filled arrows, respectively. Red lines mark targeting sequences of the crRNAs a, b, and c. **b** Scheme of the continuous evolution process with iterative cycles of mutagenesis induction. Transformed colonies of MG\*-SacB-TEM-1Δ*ung*Δ*nfi* with the plasmids pdCas9b.a.c and pSEVA221AID-T7RNAP were grown overnight (O/N) in LB with Cm and Km at 37 °C with shaking (250 r.p.m.). The next day, the cultures were diluted 1:100 in fresh medium and incubated under the same conditions for 2 h. Then anhydrotetracycline (aTc; 200 ng ml$^{-1}$) was added for induction and the cultures were incubated for 1 h. To start the second cycle of mutagenesis, the induced cultures were diluted 1:100 in new medium and grown O/N repeating the same steps as above. To monitor the mutagenic process, after every induction the cultures were washed with 1× PBS and serially diluted prior to be plated on LB agar alone and with increasing concentrations of ceftazidime (CAZ; 1, 4, and 16 μg ml$^{-1}$). The strain MG\*-SacB-TEM-1Δ*ung*Δ*nfi* with the plasmids pdCas9 and pSEVA221 was used as a control of the spontaneous mutations occurring in *TEM-1*. **c** Resistance frequency to increasing concentrations of ceftazidime (CAZ) after each cycle of the cytosine deaminase induced cultures of MG\*-SacB-TEM-1Δ*ung*Δ*nfi* pdCas9b.a.c pSEVA221AID-T7RNAP (CD). As negative control (C−), the strain MG\*-SacB-TEM-1Δ*ung*Δ*nfi* with the plasmids pdCas9 and pSEVA221 was used. The histogram shows the single values (color coded dots), means (bars), and standard errors (black lines) from three independent cultures for each strain ($n = 3$). The statistical analysis was done using a two-tailed paired *t* test. Exact *p* values (*p*) are indicated in the figure. A *p* value < 0.05 was considered significant. Source data are provided as a Source data file.

**Table 1 TEM-1 mutations in ceftazidime resistant (CAZ^R) clones.**

| Amino acid change[a] | Transition | CAZ (µg ml⁻¹) | |
|---|---|---|---|
| | | 1 | 16 |
| S4N | G to A | 0 | 1 |
| V33I | G to A | 0 | 3 |
| E104K | G to A | 0 | 12 |
| E110K | G to A | 0 | 1 |
| V119I | G to A | 0 | 1 |
| A150T | G to A | 0 | 2 |
| R164C | C to T | 4 | 0 |
| R164H | G to A | 8 | 12 |
| P174S | C to T | 0 | 1 |
| A249V | C to T | 1 | 0 |
| V262I | G to A | 0 | 1 |
| G267R | G to A | 1 | 0 |

[a]Residues numbered according to Ambler et al.[59].

**Table 2 Minimal inhibitory concentration (MIC) to ceftazidime (CAZ) of parental strain and evolved mutants.**

| Strain | MIC CAZ (µg ml⁻¹) |
|---|---|
| MG*-SacB-TEM-1Δ$ung$Δ$nfi$ | 0.25 |
| R1.10 (R164H) | 6 |
| R2.2 (E104K, R164H) | >256 |

$rpoB$ and in $URA3$ lacking $P_{T7}$. We found very high on-target vs. off-target ratios for all BD-T7RNAP fusions (ca. ≥10³). Off-target mutagenesis increased only moderately (ca. twofold) by expression of AID and TadA* fusions, and more noticiable with more active pmCDA1 and rAPOBEC1. A different situation is caused by off-target mutations in downstream sequences (relative to $P_{T7}$) of the target gene, due to the high processivity of the BD-T7RNAP fusions. To deal with this problem, the MutaT7 system included an array of at least four copies of T7 terminator downstream the target gene[21]. However this approach does not allow to define edited DNA windows within a gene. To overcome this limitation, we have used dCas9 directed with crRNAs to block transcriptional elongation, and consequently restrict the mutagenesis of BD-T7RNAP fusions to the target sequence, even within the coding sequence of a gene. Triple crRNA array/dCas9 complexes inhibited more efficiently elongation of BD-T7RNAP fusions and protected from their mutagenic activity downstream genes. When triple crRNA were directed within the coding sequence of $URA3$, mutagenesis was restricted to the gene segment proximal to $P_{T7}$, and in regions between the hybridization of sites of the crRNAs. Yet, protection was not complete and this could be improved with dCas9 variants having increased blockade activity[47].

One interesting feature of the reported method is that it could be adapted for its use in different hosts since BDs, T7RNAP, and dCas9 have been expressed in different bacteria, yeast, and mammalian cells[14,15,23,24]. For modulation of the DNA repair systems, the expression of UGI would avoid the need of delete $ung$ in host cells. Elimination of $nfi$ is not essential for TadA*-T7RNAP activity, but the modified TadA* pairs with the endogenous TadA of $E. coli$ for activity. Therefore, in hosts lacking endogenous TadA, a heterodimeric construct fusing TadA and TadA* is required[19].

We have also demonstrated that T7-DIVA system is an efficient tool for the directed evolution of genes of interest, using $TEM-1$ as proof-of-principle. After only two iterative cycles of mutagenesis, in a continuous process, we found that mutations within $TEM-1$ were accumulative given rise to CAZ^R variants at 1, 4, and 16 µg ml⁻¹ with elevated frequencies. Sequence analysis of the $TEM-1$ variants revealed the presence of different mutations that have been previously reported to provide increased levels of resistance to CAZ[42]. This highlights the potential of this in vivo mutagenesis system for the molecular evolution of proteins of interest, in a simple and continuous process. Evolution of $TEM-1$ was made upon integration in the chromosome of $E. coli$. This approach can be followed with any other gene of interest, but other alternatives exist. For instance, the T7 promoter itself can be integrated downstream of genes or operons found in the genome of $E. coli$ (or other host). The integration of multiple T7 promoters in different parts of the genome (e.g., downstream of genes involved in a metabolic pathway or in antibiotic resistance) would render in a multiplex genome editing for directed evolution of a complex cell phenotype. Alternatively, a plasmid that carries the target gene(s) and the T7 promoter could be used as shown with MutaT7[21]. The use of multicopy plasmids can ease the initial steps, but selection of a particular variant among the multiple copies of the plasmid require their isolation and retransformation, increasing manipulation and difficulting a continuous evolution.

In conclusion, the T7-DIVA system hereby documented represents a simple workflow for continuous molecular evolution of proteins of interest (e.g., antibodies and enzymes), metabolic engineering of enzymatic pathways, diversification of gene libraries, and other applications (e.g., predicting evolution of antibiotic resistance genes).

## Methods

**Bacterial strains, media, and growth conditions**. A full list of the bacterial strains used in this study can be found in Supplementary Table 1. For plasmid propagation and cloning the $E. coli$ strains used were DH10BT1R and BW25141 (for the $pir$-dependent suicide plasmid derivatives). All the strains were grown in lysogeny broth (LB) at 37 °C and shaking at 250 r.p.m.[48]. For solid medium, 1.5 % (w/v) agar was added to LB. The following antibiotics were added to the medium when needed: 50 µg ml⁻¹ kanamycin (Km), 30 µg ml⁻¹ chloramphenicol (Cm), 50 µg ml⁻¹ apramycin (Apra), and 50 µg ml⁻¹ Rif. Antibiotics were obtained from Duchefa-Biochemie. All other chemical reagents were obtained from Merck-Sigma unless indicated otherwise. For monitoring the mutagenic process, minimal medium M9 plates were used[48]. This medium contains: 1× M9 salts (1 g l⁻¹ NH₄Cl, 3 g l⁻¹ KH₂PO₄, and 6 g l⁻¹ Na₂HPO₄), 2 mM MgSO₄, 0.4 % (w/v) glucose, 0.0005 % (w/v) thiamine, and 1.6 % (w/v) agar for solidification. When required, the minimal medium was supplemented with 20 µg ml⁻¹ uracil and 250 µ ml⁻¹ FOA (Zymo Research) or 60 g l⁻¹ sucrose (counterselection with $sacB$).

**Plasmids, primers, and cloning procedures**. A list of the plasmids used in this study can be found in the Supplementary Table 2. All the primers used in this study are listed in the Supplementary Table 3. Cloning procedures were performed following standard protocols of DNA digestion with restriction enzymes and ligation[48]. All DNA constructs were sequenced by the chain-termination Sanger method (Macrogen) and the sequences are deposited in GenBank (Supplementary Table 2). The plasmid pSEVA221 (Km^R, RK2-origin)[49] was used for expression of the T7RNAP fusions under the control of the $tetR$-$P_{tetA}$ promoter[33]. The $tetR$-$P_{tetA}$ and the DNA fragments coding for the BD enzymes fused to the linker peptide (G₃S)₇ were chemically synthesized with codon optimization for expression in $E. coli$ (GeneART, ThermoFisher Scientific). The following BD enzymes were synthesized: human AID, rAPOBEC1[50], pmCDA1[51], and TadA* [19]. The plasmid pdCas9[40] was used for the constitutively expression of the catalytically dCas9, the tracrRNA, and the crRNA. The double (b.a) and triple (b.a.c and d.e.f) spacer arrays were cloned into the $Bsa$I site of pdCas9, using hybridized complimentary oligonucleotides (Supplementary Table 4, and Supplementary Figs. 10 and 11) following the one-step scheme CRATES[52]. Details of plasmid constructs and DNA synthesis are indicated in the Supplementary Information.

**Generation of the reporter strains**. The reporter strains for the mutagenesis system were derived from the reference of $E. coli$ K-12 strain MG1655[30]. The strain MG1655* with higher sensitivity to FOA was generated by an oligo-mediated allelic replacement method[11] (Supplementary Methods). The oligonucleotides used

in this method are listed in the Supplementary Table 6. All the successive modifications were done over the MG1655* genetic background. The genes *pyrF*, *ung*, and *nfi* were deleted using the plasmids pGE*pyrF*, pGE*ung*, and pGE*nfi*, respectively, by a marker-less genome edition strategy[29] (Supplementary Methods). This strategy is based on homologous recombination and resolution of the cointegrant promoted by the expression of the restriction enzyme I-*SceI* and the λRed from the plasmid pACBSR[53]. The two cassettes containing *URA3* and *TEM-1* were inserted in the *flu* locus using plasmid derivatives of pGETS[54] (Supplementary Methods).

**Induction of the mutagenesis system.** A colony of freshly transformed reporter strain with the indicated plasmid derivative of pSEVA221, was grown overnight (O/N) in LB with Km at 37 °C with shaking (250 r.p.m.). The next day, the culture was diluted 1:100 in fresh media and incubated under the same conditions for 2 h. Then aTc (200 ng ml$^{-1}$; TOKU-E) was added for induction and the cultures were incubated for 1 h. After that, a 500-μl aliquot of each culture was washed with 1× phosphate-buffered saline (PBS) and resuspended in the same volume of PBS. A series of tenfold dilutions of the cell suspension was prepared, and aliquots of 100 μl were plated in duplicates on different media: M9 + uracil; M9 + uracil and FOA; M9 + uracil and sucrose; and LB + Rif (see above for media composition).

**Flow cytometry analysis.** Levels of expression of *gfp* from induced cultures were determined by flow cytometry analysis as follows: the volume corresponding to one unit of optical density (O.D.) at 600 nm of the induced cultures was collected by centrifugation (3300 × *g*, 5 min) and resuspended in 500 μl 1× PBS. The cell suspension was diluted transferring 200 μl to a tube with 1200 μl of 1× PBS, and its fluorescence levels was determined using a Gallios FC500 flow cytometer (Beckman Coulter). Data were collected with the software CXP Cytometer version 2.2 (Beckman Coulter Inc) and analyzed with Kaluza Analysis version 2.1 (Beckman Coulter Inc). The gating strategy is shown in Supplementary Fig. 3.

**Western blot analysis.** To detect the expression of the fusion proteins in cell extracts of induced cultures, western blot analysis was performed. From induced cultures, 0.5 O.D. was collected by centrifugation 3300 × *g* 5 min, washed once with 500 μl 1× PBS, resuspended in 60 μl of PBS, and mixed with 15 μl of 5× SDS–PAGE sample buffer[55]. The samples were boiled 10 min before loaded into 8% polyacrylamide SDS gels, and electrophoresis was done using the Miniprotean III system (Bio-Rad) during 1 h 30 min at 170 V. The gels were then transferred to a polyvinylidene difluoride membrane (Immobilon-P, Millipore) by means of O/N wet transfer (Bio-Rad) 4 °C at 30 V. The membranes were blocked with PBS 0.1% (v/v) Tween 20 with 3% (w/v) skim milk powder, and successively incubated with monoclonal mouse anti-T7RNAP antibodies at a dilution of 1:10,000 (Novagen, Merck ref 70566-3, kit batch number D00143325) and POD-labeled goat anti-mouse antibodies at a dilution of 1:5000 (Sigma A2554). Membranes were developed by chemiluminiscence using the Clarity Western ECL Substrate kit (Bio-Rad) and images were acquired using a ChemiDoc Touch system (Bio-Rad).

**DNA sequencing and analysis.** To identify the mutations that caused Rif resistance, the RRDR[38] of the *rpoB* genes, a DNA fragment of 773 bp containing the RRDR was amplified using the pair of primers F_rpoB/R_rpoB (Supplementary Table 7), with the GoTaq Flexi DNA polymerase (Promega) by colony PCR following manufacturer's instructions. The resulting amplicon was sequenced by the Sanger chain-termination method (Macrogen) using the same primers. The resulting two reads per colony were mapped against the *rpoB* reference sequence to detect variants, using the program SeqMan Pro (DNAstar).

To determine the DNA sequence of the *URA3* alleles in the FOA resistant colonies, the same strategy as with the RRDRs was followed. In this case, 1191 bp amplicons were obtained with the pair of primers F_GFPseq/R_T0ter (Supplementary Table 7), and subsequently sequenced by the Sanger chain-termination method (Macrogen), using the same primers. Variants of the Ptac-*URA3*-P$_{T7}$ reference sequence (942 bp) were detected with the program SeqMan Pro (DNAstar).

For deeper analysis of the variations in *URA3*, NGS sequencing analysis of a 200 bp region was performed. To do this, genomic DNA was extracted from ~4.5 ml of each induced culture (~5 × 10$^9$ bacteria) using the GNOME DNA Kit (MP Biomedicals). One hundred ng of total genomic DNA was used as template in a PCR reaction to amplify 284 bp of *URA3* with the pair of primers F_CS1_URA3/R_CS2_URA3 that includes Illumina tags CS1 and CS2 (Supplementary Table 7). The DNA amplification was carried out in 50 μl reactions using Herculase II Fusion DNA polymerase (Agilent Technologies) following manufacturer's instructions. The amplicons were sent to the Genomic Unit of the Madrid Scientific Park to be sequenced by the NGS platform Illumina Miseq with paired-end (length >2 × 300 bp) to acquire ~1,000,000 reads per sample. The raw data were collected with BaseSpace Sequence Hub version 5.28 software (Illumina, Inc.). These reads were processed with the program Bbmap version 28.36[56] for merging the paired-end reads. The resulting merged files were pilep up against the reference sequence using the program Samtools version 1.9[57] and the variants were obtained with the program VarScan version 2.4.3[58] with the following parameters:–min-coverage 1–min-reads2 1–min-avg-qual 40–min-var-freq 0.000001–*p* value 0.99. The sequences of the oligonucleotides used for amplification were discarded from the analysis since they may contain variations due to chemical synthesis.

**TEM-1 evolution and characterization of the evolved mutants.** The gene *TEM-1* was evolved by subjecting the strain MG*-SacB-TEM-1Δ*ung*Δ*nfi* to two mutagenesis cycles with AID-T7RNAP as follows. Three freshly transformed colonies of MG*-SacB-TEM-1Δ*ung*Δ*nfi* with the plasmids pdCas9b.a.c and pSEVA221AID-T7RNAP were grown O/N in LB, with Cm and Km at 37 °C with shaking (250 r.p.m.). The next day, the cultures were diluted 1:100 in fresh medium and incubated under the same conditions for 2 h. Then aTc (200 ng ml$^{-1}$; TOKU-E) was added for induction and the cultures were incubated for 1 h. To start the second cycle of mutagenesis, the induced cultures were diluted 1:100 in new medium and grown O/N repeting the same steps as above. To monitor the mutagenic process, after every induction the cultures were washed with 1× PBS and serially diluted prior to be plated on LB agar alone and with increasing concentrations of CAZ (1, 4, and 16 μg ml$^{-1}$). The strain MG*-SacB-TEM-1Δ*ung*Δ*nfi* with the plasmids pdCas9 and pSEVA221 was used as a control of the spontaneous mutations occurring in *TEM-1*.

The sequences of the *TEM-1* genes from selected CAZ-resistant colonies were determined by PCR amplification with the primers F_GFPseq/R_T0ter (Supplementary Table 7), and subsequent Sanger sequencing method, as described above. The MIC to CAZ of parental strain MG*-SacB-TEM-1Δ*ung*Δ*nfi* and evolved mutants was assessed with the ETEST method (Liofilchem), following manufacturer's instructions.

**Statistics.** Means and standard errors of experimental values were calculated using Prism 8.0 (GraphPad software Inc). Statistical analyses comparing groups in pairs were performed using Mann–Whitney test (Figs. 2d and 4e, f, and Supplementary Figs. 6 and 7), two-tailed Student's *t* test (Figs. 3d, f and 5d) and paired *t* test (Fig. 7c) from at least three independent experiments. A value of *p* < 0.05 was considered significant.

**Reporting summary.** Further information on research design is available in the Nature Research Reporting Summary linked to this article.

## Data availability

Data that support the findings of this work can be found in the main manuscript and in the Supplementary Information. Figures with associated raw data are Figs. 2d, e, 3d–f, 4a–f, 5c, d, 6b and 7c; the Supplementary Figs. 2d–f, 4, 6, 7, 8a, b and 9; the uncropped blots from the Figs. 2b and 3b, and Supplementary Fig. 2b with the information about the antibodies used and the working dilution; and the original uncropped images from the Supplementary Fig. 1b. The sequences of the constructs built for this study are deposited in GenBank with the accession numbers and their hyperlinks listed in Supplementary Table 2. Sequencing data from highthrough-put DNA sequencing experiments are deposited in Sequencing Read Archive (SRA) in the Bioproject ID PRJNA675288. Materials and additional data are available from the corresponding author upon request. Source data are provided with this paper.

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

## Acknowledgements

We thank Dr. David Bickard (Institute Pasteur, France) and Drs. Esteban Martínez, Yamal Al-ramahi, and Tomás Aparicio (CNB-CSIC) for providing some materials used in this work. The excellent technical work in massive DNA sequencing of the Genomic Unit of Parque Científico de Madrid is greatly appreciated. We are grateful to Dr. Alejandra Bernardini (Hospital 12 de Octubre Madrid, Spain) for assistance with massive DNA sequencing data analysis. This work was supported by the Grants BIO2017-89081-R (Agencia Española de Investigación AEI/MICIU/FEDER, EU) to L.A.F. and ERC-2012-ADG_20120314– (European Research Council,EU) to V.d.L.

## Author contributions

L.A.F., M.M., and V.d.L. conceived the study. B.A. and L.A.F. designed the experiments and analyzed the results. B.A. performed the experiments. All authors interpreted the data. B.A. and L.A.F. wrote the initial manuscript and prepared figures. All the authors revised and approved the final manuscript.

## Competing interests

The authors declare no competing interests.
