## [Peer Review File · Nature Communications]

Reviewers' Comments:

Reviewer #1:

Remarks to the Author:

The authors present a novel strategy to diversify the nucleotide sequence of a target gene in *E. coli* by fusing the T7 RNA polymerase to a base deaminase. The fusion specifically transcribes a target gene neighboring a T7 promoter, and the base deaminase causes nucleotide transitions as the fusion proceeds down the length of the gene. In this way, the authors suggest they can diversify a greater length of sequence in *E. coli* than is possible with other *in vivo* mutagenesis techniques. Further, they suggest that dCas9 can be targeted to block the elongation of this mutagenic fusion protein at a specified location, such that only the sequence between the T7 promoter and the dCas9 binding site will be mutagenized.

The authors point out that a major benefit of *in vivo* mutagenesis is that it is not necessary to clone and deliver libraries of variants to cells. Here, the authors engineer a strain with a UNG and endonuclease V deletion (to improve deaminase efficiency), and then deliver a target gene with a T7 promoter, and dCas9 with a guide array. In other words, strain engineering, cloning, and transformation are required. They achieve mutation frequencies ranging from 1 – 14 mutations per kilobase. They report off-target mutation with some of these fusions, up to 1 in 1,000,000 colonies gaining a mutation in the particular off-target gene *rpoB* that is checked with a rifampicin resistance assay, representing a 5-fold increase in *rpoB* mutation over a negative control. The technique may have some advantages but its implementation is not simple, as with most other mutagenesis techniques.

Major Comments

1. Fig 1iii suggests the dCas9 can be programmed to stop the mutations at a specific location within the target gene, by choosing a specific guide RNA. This is an interesting use of dCas9 and is supported by the phenotype data in Fig 6D. Can the authors include sequencing data to further support this concept?
2. Do the authors have data with a better negative control guide array for the CRISPR experiment – such as guides that target elsewhere in the genome? Is the current negative control a 'no spacer' array?
3. The authors suggest exciting applications such as integrating the T7 promoter at an endogenous gene or using the system for continuous directed evolution. The data in the paper is entirely derived from mutagenesis of a reporter gene. Do the authors have data to demonstrate some of the more impactful applications that they mention?

Minor Comments

1. Dots instead of bars could help the clarity of the log mutation frequency plots.
2. Would the authors consider sharing the plasmids via a repository (e.g. Addgene) to ease adoption of their method?
3. Perhaps the NGS data could be brought out of the supplement and into Figure 5?

Reviewer #2:

Remarks to the Author:

The authors create a series of BD-T7RNAP fusions and extensively study their ability to achieve targeted mutagenesis *in vivo* in the context of various strain modifications. The authors demonstrate many interesting additions to BD-T7RNAPs-driven targeted mutagenesis, compared to earlier work by Moore et al. (currently citation 59), especially the dCas9 blockade that should prove useful to limit the boundaries of mutagenesis to one side of a gene. However, in order for this work to reach the significance and novelty needed for publication in *Nature Communications*, it is my opinion that the authors should make two major additions.

First, although the authors claim that their system can be used for continuous evolution, this is not shown at all. The mutation assays are done by inducing the BD-T7RNAP for a short period and then plating cells to look for mutants. Therefore what is shown is just one round of mutagenesis and selection. The authors need to design a model system where we can observe the sequential discovery and fixation of beneficial mutations over a longer process for this mutagenesis technique to enter the possibility of continuous evolution. Such a system might involve continuous induction of the mutagenesis machinery or regular pulses of induction, combined with tracking of mutational accumulation over time. I would also suggest an actual directed evolution experiment to show this, where we would like to see that the system is reliable enough to carry out mutagenesis over a period of time relevant to evolve a new function from a protein. The state of the art in the field of continuous evolution are experiments demonstrating the evolution of new activity through many mutations that accumulate and fix over time.

Second, I believe that at least conceptually, the only main innovation in this work over Moore et al. is the dCas9 blockade. However, this blockade strategy needs to be improved. First, a 10- to 14-fold drop in mutation seems too low for isolation of the mutagenesis region, since mutation rate elevations are in the hundreds to thousands. Second, the blockade strategy would be useful only if one can do it within a gene. This is because if not within a gene, a strong T7 terminator can be used, which is how Moore et al. limits mutation. I suggest the authors show that if we do dCas9 blockade within, for example, URA3, we see mutations only in a limited region. (This should work as dCas9 repression is strand specific.) I also suggest the authors improve the blockade efficiency. Only if both of these are done will the blockade strategy preferable over the obvious one of using a T7 terminator (or an array of them).

Besides the main points, I have some other comments:

1) In my opinion, the authors should prominently discuss Moore et al. in the introduction rather than citing it as citation 59. It is conceptually similar so it should be treated as relevant prior work to the system presented. If the authors feel that Moore et al. is conceptually distinct for reasons that I have not noticed, then the authors should explain the distinctions in the introduction, comparing to Moore et al.

2) In the discussion of yeast in vivo continuous evolution systems, the retrotransposon system from Crook et al. is mentioned. The more recent orthogonal DNA replication (Ravikumar et al., Cell, 2018) should also be mentioned.

3) I gather from the methods that the BD-T7RNAP is under an inducible promoter. This should be made clear in the main text. The implication of induced rather than constitutive mutagenesis could be discussed.

4) The rpoB mutants that arise from off-target activity should be sequenced so that we can see if they are indeed the mutations expected from deaminase, supporting the validity of rpoB as an off-target mutagenesis sensor. If they are not deaminase-consistent mutants, then we would conclude that rpoB is insensitive to the off-target mutation activity we want to detect and another locus should be chosen.

5) It seems the authors are achieving a rather high mutation rate as they only have to induce for an hour to see many URA3 mutants. This is quite exciting and provides the opportunity to do a detailed time course where we can observe the mutation accumulation rate to get a better measure of mutation rate than the mutant frequencies presented. Mutant frequencies are difficult to interpret and compare across technologies whereas mutation rates are better for comparison and more intuitive to understand.

6) Technically, it would be better to do mutant frequency measurements in fluctuation analysis

format (following luria delbruck) since there is cell division between the start of the mutagenesis pulse to selection for mutants. When there is cell division during mutagenesis, then mutant frequency across different experiments does not follow a normal distribution around mutation rate, so statistical analysis has to be adjusted. Since the mutagenesis induction is short here, I'm quite certain this is likely not a problem with the data presented (although it could still upset mutation rate measurement aficionados). However, it is a point that the authors should think about if they do some of the experiments I suggested earlier involving longer mutation times where more cell division will occur.

Referee #1

Major Comments

Q1. Fig 1iii suggests the dCas9 can be programmed to stop the mutations at a specific location within the target gene, by choosing a specific guide RNA. This is an interesting use of dCas9 and is supported by the phenotype data in Fig 6D. Can the authors include sequencing data to further support this concept?

A1. New experiments have been done to demonstrate this concept. New Figure 6 shows sequencing data demonstrating that triple crRNA array targeted toward the middle of URA3 protects downstream regions within this gene from the action of BD-T7RNAP. This is incorporated in Results page 11, lines 312-325, as follows:

" To evaluate whether dCas9 can be used to protect a particular region within a gene, we designed a new triple crRNA array (d.e.f) targeting URA3 (Fig. 6a). Cultures of MG⁻-SacB-URA3 Δ ung Δ nfi strain expressing AID-T7RNAP and dCas9 with crRNAd.e.f (targeting URA3), or crRNAb.a.c (targeting *gfp*) as a control, were induced and the URA3 alleles from 30 FOA^R colonies from each culture were sequenced. As expected, mutations in FOA^R clones expressing crRNAb.a.c (targeting *gfp*) were found distributed all along URA3, whereas mutations in FOA^R clones expressing crRNAd.e.f (targeting URA3) were only found in the gene segment between the recognition sites of the crRNAs and P_{T7} (Fig. 6b). Interestingly, albeit mutations in URA3 were not detected more distal than the hybridization site of crRNA d, we found mutations in regions between the hybridization sites of crRNAs d to e and e to f (Fig. 6b), as well as in the URA3 segment proximal to P_{T7}. Collectively, these results demonstrate that targeted dCas9 blockade with a triple crRNA array can be used to concentrate the mutagenesis activity of BD-T7RNAP fusions to a specific target gene or gene segment, reducing the mutagenesis of downstream DNA regions."

Q2. Do the authors have data with a better negative control guide array for the CRISPR experiment – such as guides that target elsewhere in the genome? Is the current negative control a 'no spacer' array?

A2. In the experiment of New Figure 6 we have incorporated a control of crRNA array to *gfp* and in this case sequencing shows that URA3 is not protected.

Q2. The authors suggest exciting applications such as integrating the T7 promoter at an endogenous gene or using the system for continuous directed evolution. The data in the

paper is entirely derived from mutagenesis of a reporter gene. Do the authors have data to demonstrate some of the more impactful applications that they mention?

A2. In the revised manuscript we have applied our system for the continuous directed evolution of TEM1- β -lactamase, isolating in just two cycles of continuous growth and induction, multiple TEM-1 variants with increase resistance to third generation cephalosporin ceftazidime (CAZ). These results are described in Results section " *Fast directed evolution of the TEM-1 gene*" pages 12-13, new Fig. 7 and Tables 1 and 2.

Minor Comments

1. *Dots instead of bars could help the clarity of the log mutation frequency plots.*

A: All Figures are remade with Prism 8 software that allow showing dots of individual experiments used for mean and statistics.

2. *Would the authors consider sharing the plasmids via a repository (e.g. Addgene) to ease adoption of their method?*

A: Yes, plasmids will be freely available upon request to the corresponding author and we will deposit them in Addgene once the paper is published. The complete sequences of key plasmids are already deposited in GeneBank (Supplementary Table 2)

3. *Perhaps the NGS data could be brought out of the supplement and into Figure 5?*

A: As suggested, new Fig. 4 (old Fig.5) now contains data of massive sequencing (panels 4e and 4f) that were previously part of Supplementary Figs. 4 and 5 (current Supp. Figs. 5 and 6 contain the rest of massive sequencing data).

Referee #2

Main Comments

Q1. First, although the authors claim that their system can be used for continuous evolution, this is not shown at all....

A1. As for Referee #1, A2. In the revised manuscript we have applied our system for the continuous directed evolution of TEM1- β -lactamase, isolating in just two cycles of continuous growth and induction, multiple TEM-1 variants with increase resistance to third generation cephalosporin ceftazidime (CAZ). These results are described in Results section " *Fast directed evolution of the TEM-1 gene*" pages 12-13, new Fig. 7 and Tables 1 and 2.

Q2. Second, the blockade strategy would be useful only if one can do it within a gene. This is because if not within a gene, a strong T7 terminator can be used, which is how Moore et al. limits mutation. I suggest the authors show that if we do dCas9 blockade within, for example, URA3, we see mutations only in a limited region.

A2: As for Referee #1, A1: A1. New experiments have been done to demonstrate this concept. New Figure 6 shows sequencing data demonstrating that triple crRNA array targeted toward the middle of URA3 protects downstream regions within this gene from the action of BD-T7RNAP. This is incorporated in Results page 11, lines 312-325.

Other Comments

1) *In my opinion, the authors should prominently discuss Moore et al. in the introduction rather than citing it as citation 59. It is conceptually similar so it should be treated as relevant prior work to the system presented. If the authors feel that Moore et al. is conceptually distinct for reasons that I have not noticed, then the authors should explain the distinctions in the introduction, comparing to Moore et al.*

A: Yes, we agree with the referee and the revised version now discuss the work of Moore et al. (now ref. 21) and also a more recent work in mammalian cells (Chen et al., ref. 22) in various sites, including Introduction (lines 112-120):

Two recent independent studies have also reported the mutagenic activity of CD fusions to T7RNAP, in *E. coli*²¹ and in mammalian cells²². Moore *et al.* described a chimeric fusion between rAPOBEC1 and T7RNAP (MutaT7) that in *E. coli* introduces mutations within a plasmid DNA segment downstream of P_{T7}. Due to the high processivity of this fusion, an array of at least 4 copies of the T7 terminator were needed downstream of the target gene to restrict the mutagenesis²¹. The study in mammalian cells uses fusions of rAPOBEC1 or a hyperactive mutant AID to T7RNAP. Mutations in a DNA segment of at least 2000 bp were reported, but no brake system was included to delimit the edited window²².

and Discussion (lines 403-405):

Also interesting is that the mutagenic activity of all BD-T7RNAP fusions is biased towards the non-template strand. This phenomenon is less pronounced for AID fusion, but it is found in all of them. This observation was also reported for rAPOBEC1 in MutaT7²¹.

Discussion (lines 415-419)

A different situation is caused by off-target mutations in downstream sequences (relative to P_{T7}) of the target gene, due to the high processivity of the BD-T7RNAP fusions. To deal with this problem, the MutaT7 system included an array of at least 4 copies of T7 terminator downstream the target gene²¹.

Discussion (lines 450-451)

Alternatively, a plasmid that carries the target gene(s) and the T7 promoter could be used as shown with MutaT7²¹.

2) *In the discussion of yeast in vivo continuous evolution systems, the retrotransposon system from Crook et al. is mentioned. The more recent orthogonal DNA replication (Ravikumar et al., Cell, 2018) should also be mentioned.*

A: This new reference Ravikumar *et al.* (13) is described in the Introduction, along with that of Crook et al. (12)

Introduction (lines 80-86)

In yeast, generation of variability can be achieved by cloning the target DNA segments up to 5 kb in retrotransposons having an error-prone retrotranscriptase¹². Recently, an *in vivo* mutagenesis system, named OrthoRep, uses a highly-error-prone orthogonal DNA polymerase and a linear plasmid pair that allows the directed evolution of DNA segments of at

least 18 kb¹³. However, directed evolution of gene segments or domains is challenging since the orthogonal DNA polymerase replicates the whole plasmid.

3) I gather from the methods that the BD-T7RNAP is under an inducible promoter. This should be made clear in the main text. The implication of induced rather than constitutive mutagenesis could be discussed.

A: The inducibility of the system was described in the Results and Discussion as well, albeit maybe the referee did not notice it. We have tried to further clarify this along the text indicating that cultures were grown and induced.

Results (lines 160-162)

Native T7RNAP and these fusions were cloned under the control of the tetracycline-inducible promoter (TetR-PtetA)³³ in a low copy-number plasmid (pSEVA221)³⁴

Results (lines 167-169)

MG*-URA3 Δ ung strains carrying plasmids encoding native T7RNAP, AID-T7RNAP, pmCDA1-T7RNAP, rAPOBEC1-T7RNAP, or pSEVA221 (negative control), were grown and induced with anhydrotetracycline (aTc).

Results (lines 250-252)

For deeper analysis, a 284 bp PCR fragment of the URA3 gene was amplified from an induced culture of MG*-URA3 Δ ung expressing AID-T7RNAP without FOA selection, ..

Results (lines 297-298)

After growth and aTc induction, we determined that GFP expression by AID-T7RNAP...
etc.

Discussion (lines 378-381)

Expression of the BD-T7RNAP fusions is inducible with aTc. An inducible expression of the mutagenic enzyme reduces potential toxicities associated to the constitutive expression of BD and/or T7RNAP, and it is valuable to control the mutagenic process.

4)The rpoB mutants that arise from off-target activity should be sequenced so that we can see if they are indeed the mutations expected from deaminase, supporting the validity of rpoB as an off-target mutagenesis sensor. If they are not deaminase-consistent mutants, then we would conclude that rpoB is insensitive to the off-target mutation activity we want to detect and another locus should be chosen.

A: We add new experimental data and Figure (Supplementary Fig. 7) showing mutations in *rpoB* from Rif^R-colonies, including transitions C>T and A>G, thus demonstrating that *rpoB* is a good reporter to evaluate off-target activity of BD fusions. This is explained in the Results (lines 263-272)

We also confirmed the adequacy of the Rif^R phenotype as reporter of the off-target activity of CD and AD fusions. The Rif^R phenotype is mostly caused by *rpoB* mutations in a region between amino acid 507 and 687 of the β -subunit of *E. coli* RNAP known as RIF-resistant determining region (RRDR)³⁸. We sequenced the *rpoB*-RRDR segment in 30 Rif^R colonies from strains MG^{*}-URA3 Δ *ung* and MG^{*}-URA3 Δ *nfi* carrying pSEVA221. Among other mutations, transitions C to T and G to A were readily found in *rpoB*-RRDR from Rif^R MG^{*}-URA3 Δ *ung* (Supplementary Fig. 7a). Similarly, Rif^R MG^{*}-URA3 Δ *nfi* contained diverse transitions A to G and T to C in *rpoB*-RRDR (Supplementary Fig. 7b). Consequently, different bases within *rpoB*-RRDR that are sensitive to CD and AD mutagenesis can generate Rif^R mutants, validating *rpoB* to assess the off-target activity of BDs.

5) It seems the authors are achieving a rather high mutation rate as they only have to induce for an hour to see many URA3 mutants. This is quite exciting and provides the opportunity to do a detailed time course where we can observe the mutation accumulation rate to get a better measure of mutation rate than the mutant frequencies presented. Mutant frequencies are difficult to interpret and compare across technologies whereas mutation rates are better for comparison and more intuitive to understand.

A: Induction is done for one hour in all experiments but the mutagenesis frequencies described in most of the Figures reflect the growth of colonies on plates O/N, making difficult to assess a proper "mutation rate". Trying to address this, we performed massive sequencing of induced cultures without selection (DNA isolated immediately after induction) which show "rates" as number of mutations in a 200 bp region in 1 million sequence reads, which average $\sim 0.5 \times 10^{-3}$ for AID and less for TadA fusions (Fig. 4f and 4e).

6) Technically, it would be better to do mutant frequency measurements in fluctuation analysis format (following luria delbruck) since there is cell division between the start of the mutagenesis pulse to selection for mutants. When there is cell division during mutagenesis, then mutant frequency across different experiments does not follow a normal distribution around mutation rate, so statistical analysis has to be adjusted. Since the mutagenesis induction is short here, I'm quite certain this is likely not a problem with the data presented (although it could still upset mutation rate measurement aficionados). However, it is a point that the authors should think about if they do some of the experiments I suggested earlier involving longer mutation times where more cell division will occur.

A: We appreciate the comment and will consider the suggested experiments for future work.

LAF

Reviewers' Comments:

Reviewer #2:

Remarks to the Author:

The manuscript is much improved and demonstrates the main claims. Although the directed evolution experiment on beta-lactamase is still quite a modest demonstration of the potential of the system, I believe the manuscript is acceptable for publication.

Point-by-point response to Referees of Álvarez et al. NCOMMS-19-38437
(revised version)

Referee #2

The manuscript is much improved and demonstrates the main claims. Although the directed evolution experiment on beta-lactamase is still quite a modest demonstration of the potential of the system, I believe the manuscript is acceptable for publication.

A1. We thank the reviewers for their positive opinion on our work. We agree with the reviewer #2 that the directed evolution of beta-lactamase is a modest demonstration of the potential of the system. Indeed, we clearly state on the text that this is a proof-of-principle (Results: "*To test this in vivo mutagenesis system in a directed evolution process, we chose the antibiotic resistance gene TEM-1 as proof-of-principle*"). We also recognize in the text the potential of the system for different applications (Discussion: "*...represents a simple workflow for continuous molecular evolution of proteins of interest (e.g. antibodies, enzymes), metabolic engineering of enzymatic pathways, diversification of gene libraries, and other applications (e.g. predicting evolution of antibiotic resistance genes)*"). Nonetheless, specific use of the system for all these applications is beyond the scope of this paper.

LAF